# Unequivocal identification of two-bond heteronuclear correlations in natural products at nanomole scale by i-HMBC

Yunyi Wang[1], Aili Fan[2], Ryan D. Cohen [1], Guilherme Dal Poggetto [1], Zheng Huang [3], Haifeng Yang [3], Gary E. Martin[4], Edward C. Sherer [1], Mikhail Reibarkh [1] ✉ & Xiao Wang [1] ✉

HMBC is an essential NMR experiment for determining multiple bond heteronuclear correlations in small to medium-sized organic molecules, including natural products, yet its major limitation is the inability to differentiate two-bond from longer-range correlations. There have been several attempts to address this issue, but all reported approaches suffer various drawbacks, such as restricted utility and poor sensitivity. Here we present a sensitive and universal methodology to identify two-bond HMBC correlations using isotope shifts, referred to as i-HMBC (isotope shift detection HMBC). Experimental utility was demonstrated at the sub-milligram / nanomole scale with only a few hours of acquisition time required for structure elucidation of several complex proton-deficient natural products, which could not be fully elucidated by conventional 2D NMR experiments. Because i-HMBC overcomes the key limitation of HMBC without significant reduction in sensitivity or performance, i-HMBC can be used as a complement to HMBC when unambiguous identifications of two-bond correlations are needed.

Since being introduced in 1986[1], HMBC quickly became a pivotal NMR experiment for structure elucidation of small molecules, including a diverse array of natural products, due to the enhanced ability of the experiment to detect long-range correlations between NMR active nuclei. HMBC can routinely detect correlations between $^1$H and $^{13}$C nuclei separated by two to four bonds, and occasionally even five or six bonds[2–5]. However, a fundamental limitation is the inability to differentiate two-bond correlations ($^2J_{CH}$) from other longer-range correlations, particularly $^3J_{CH}$. Reliable differentiation of two- and three-bond correlations in HMBC is not possible due to overlapping ranges of the magnitudes of $^2J_{CH}$ and $^3J_{CH}$ coupling constants: from 0 to 8 Hz. Typically, carbon-carbon bond connections in a $C_aH$-$C_bH$ spin system have been established by a combination of COSY and HSQC experiments, or other COSY-based experiments such as $^2J,^3J$-HMBC or H2BC[6–9]. However, this limitation is problematic for proton-deficient

compounds with a preponderance of quaternary carbons or heteroatoms, and has led to frequent "guesswork" to elucidate the carbon skeleton from HMBC correlations, due to the inability to differentiate the number of bonds between atoms. This tedious and error-prone workflow has frequently yielded inconclusive or misassigned structures, leading to the heuristic "Crews rule" – a low ratio of H/C (<1:1) could complicate or even prevent unequivocal structure elucidation using the conventional suite of 2D NMR experiments (e.g., COSY/TOCSY, HSQC, HMBC, and NOESY/ROESY)[10–13].

Recent advancements of NMR techniques have led to the unambiguous structure elucidation of proton-deficient compounds that meet Crews' rule. Particularly, the atom connectivity of a $C_aH$-$C_b$ spin system, where $C_b$ is a quaternary carbon, can be directly determined using the 1,1-ADEQUATE experiment, which utilizes $^1J_{CH}$ and $^1J_{CC}$[14–19]. More recently, the SEA-XLOC pulse sequence was developed to

[1]Analytical Research & Development, Merck & Co. Inc, Rahway, NJ 07065, USA. [2]State Key Laboratory of Natural and Biomimetic Drugs, School of Pharmaceutical Sciences, Peking University, Beijing 100191, P. R. China. [3]Process Research & Development, Merck & Co. Inc, Rahway, NJ 07065, USA. [4]Department of Chemistry and Biochemistry, Seton Hall University, South Orange, NJ 07079, USA. ✉e-mail: mikhail_reibarkh@merck.com; xiao.wang1@merck.com

identify two-bond proton-carbon correlations based on the observation that most $^2J_{CH}$ have a negative sign[20–23]. However, both methods have notable limitations. 1,1-ADEQUATE, which relies on magnetization transfer through two consecutive $^{13}C$ nuclei at natural abundance, is inherently about two orders of magnitude less sensitive than HMBC-based approaches for compounds without $^{13}C$ labeling. Thus, it usually requires several milligrams of material in combination with the use of a cryogenically-cooled NMR probe. Alternatively, the SEA-XLOC experiment, while considerably more sensitive than 1,1-ADEQUATE, assumes that $^2J_{CH}$ is negative, which is not always valid especially in molecules heavily substituted by atoms with high electronegativity such as oxygen[24]. As illustrated in Fig. 1, these limitations could be overcome with a straightforward and highly sensitive HMBC-derived method to detect $^n\Delta^1H(^{13/12}C)$ isotope shifts.

Here, we present such a method that enables identification of two-bond correlations with high fidelity, and which we refer to as i-HMBC for isotope shift detection HMBC. We then show the effectiveness of the technique at sub-mg (or nanomole) scale for the analysis of the highly complex, proton-deficient natural product, homodimericin B.

## Results and discussion

### Accurate measurement of $^{2-n}\Delta^1H(^{13/12}C)$ isotope shift by i-HMBC

Considering that the natural isotopic abundance of NMR active $^{13}C$ is approximately 1.1%, the two-bond ($H_a$ to $C_b$) and three-bond ($H_a$ to $C_c$) HMBC correlations in a $C_aH\text{-}C_b\text{-}C_c$ spin system are principally comprised of the isotopomers, $^{12}C_aH\text{-}^{13}C_b\text{-}^{12}C_c$ and $^{12}C_aH\text{-}^{12}C_b\text{-}^{13}C_c$, respectively, while the $H_a$ signal in the $^1H$ NMR spectrum is dominated by the $^{12}C_aH\text{-}^{12}C_b\text{-}^{12}C_c$ isotopomer. Comparing the chemical shifts of $H_a$ in the HMBC correlation versus the $^1H$ spectrum, a two-bond isotope shift $^2\Delta^1H_a(^{13/12}C)$ and a three-bond isotope shift $^3\Delta^1H_a(^{13/12}C)$ would be expected for the two- and three-bond HMBC correlations, respectively. Since the origin of the isotope shift is closely related to the anharmonic vibration of chemical bonds, one-bond isotope shifts are expected to be significantly larger than two-bond isotope shifts, which should be correspondingly larger than longer-range isotope shifts[25–28]. Consequently, if the difference between the two- and three-bond (or longer-range) isotope shifts, $^{2-n}\Delta^1H(^{13/12}C)$, is detectable in an HMBC spectrum, then the key two-bond proton-carbon correlation can be identified to establish the corresponding carbon-carbon connectivity analogous to what would be obtained in a 1,1-ADEQUATE spectrum albeit with much higher sensitivity.

There were two major challenges for this approach. (1) While long-range $^{13/12}C$ isotope shifts for $^{19}F$, $^n\Delta^{19}F(^{13/12}C)$, are large and therefore well studied[29–34], the analogous $^n\Delta^1H(^{13/12}C)$ for $^1H$ are quite small with only scarce reports to date[35–38]. Notably, prior to this work, the only reported method capable of accurate measurement of $^2\Delta^1H(^{13/12}C)$ related to quaternary carbon relied on spin noise detection[38], which requires an almost neat sample and overnight data acquisition. Such constraints are completely impractical for structure elucidation of natural products, synthetic impurities, and even products from small scale reactions. In addition, accurate measurement of $^n\Delta^1H(^{13/12}C)$ ($n \geq 3$) has not been achieved to date. (2) Recently several 2D NMR methods for measuring 5–15 ppb isotope shift of $^1\Delta^{13}C(^{37/35}Cl)$ and $^1\Delta^{15}N(^{37/35}Cl)$ were reported[39–42]. The experiments relied on very narrow band-selective refocusing of the selected carbon nucleus to achieve ~3 ppb/pt resolution on the indirect (F1) dimension. In comparison, routine HMBC experimental parameter settings afford low resolution on the direct (F2) dimension (5–13 ppb/pt), so it was unknown if HMBC could detect sub ppb level isotope shift that other higher resolution NMR experiments have struggled with.

Thus, we first aimed to develop a method to accurately measure the difference between the two- and three-bond isotope shifts, $^{2-3}\Delta^1H(^{13/12}C)$, using $^{13}C$ selectively labeled ethyl acetate (EtOAc) as a model substrate so the measured values could be used as a reference for further method development. EtOAc was synthesized with selective $^{13}C$ labeling at either the C1 carbonyl position (**1a**) or the C2′ methyl group (**1b**); their respective $^1H$ NMR spectra were then acquired (Fig. 2). Compared with the unlabeled EtOAc, the methylene H1′ of **1a** should show a three-bond $^{13}C$ isotope shift, $^3\Delta^1H(^{13/12}C)$, and H1′ of **1b** should show a two-bond $^{13}C$ isotope shift, $^2\Delta^1H(^{13/12}C)$. The difference in H1′ chemical shifts between **1b** and **1a** is the difference of the two- and three-bond $^{13}C$ isotope shifts $^{2-3}\Delta^1H(^{13/12}C)$. Experimentally, H1′ of **1a** was found to be a quartet of doublets (qd), with 7.1 Hz $^3J_{HH}$ to H2′ and 3.1 Hz $^3J_{CH}$ to C1. The H1′ of **1b** was also a qd, with 7.1 Hz $^3J_{HH}$ but 2.6 Hz $^2J_{CH}$ to C2′ (Figs. 2a, b). Direct comparison of the chemical shifts between the two qd peaks should simply yield the desired $^{2-3}\Delta^1H(^{13/12}C)$; however, this measurement was more challenging than initially anticipated. Errors in chemical shift measurements caused by referencing, minor peak distortions, and subtle concentration differences were found to be approximately the same magnitude as the isotope shift difference. When both $^1H$ spectra of **1a** and **1b** were referenced to the residual $CHCl_3$ peak, $^{2-3}\Delta^1H(^{13/12}C)$ was measured to be −0.11 ppb, while when both spectra were referenced to the TMS peak, $^{2-3}\Delta^1H(^{13/12}C)$ was measured to be −0.21 ppb. To minimize the spectral inconsistency caused by shimming and referencing, equimolar mixtures of **1a** and **1b** were prepared. However, the isotope shift difference was so small that peak

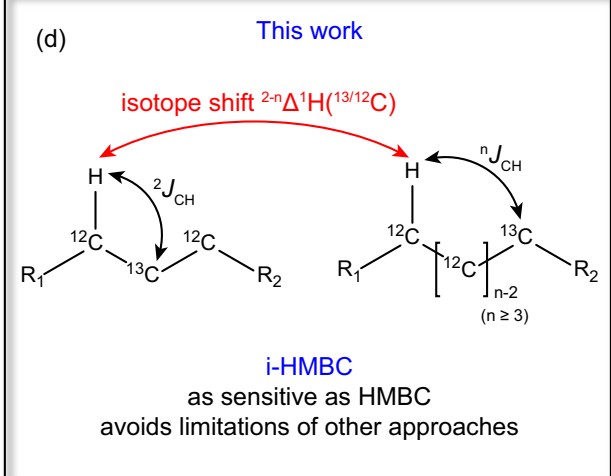

**Fig. 1 | Overview of methods capable of identification of two-bond $^1H\text{-}^{13}C$ correlations. a** COSY/$^2J$, $^3J$-HMBC/H2BC, **b** 1,1-ADEQUATE, **c** SEA-XLOC and **d** this work: i-HMBC.

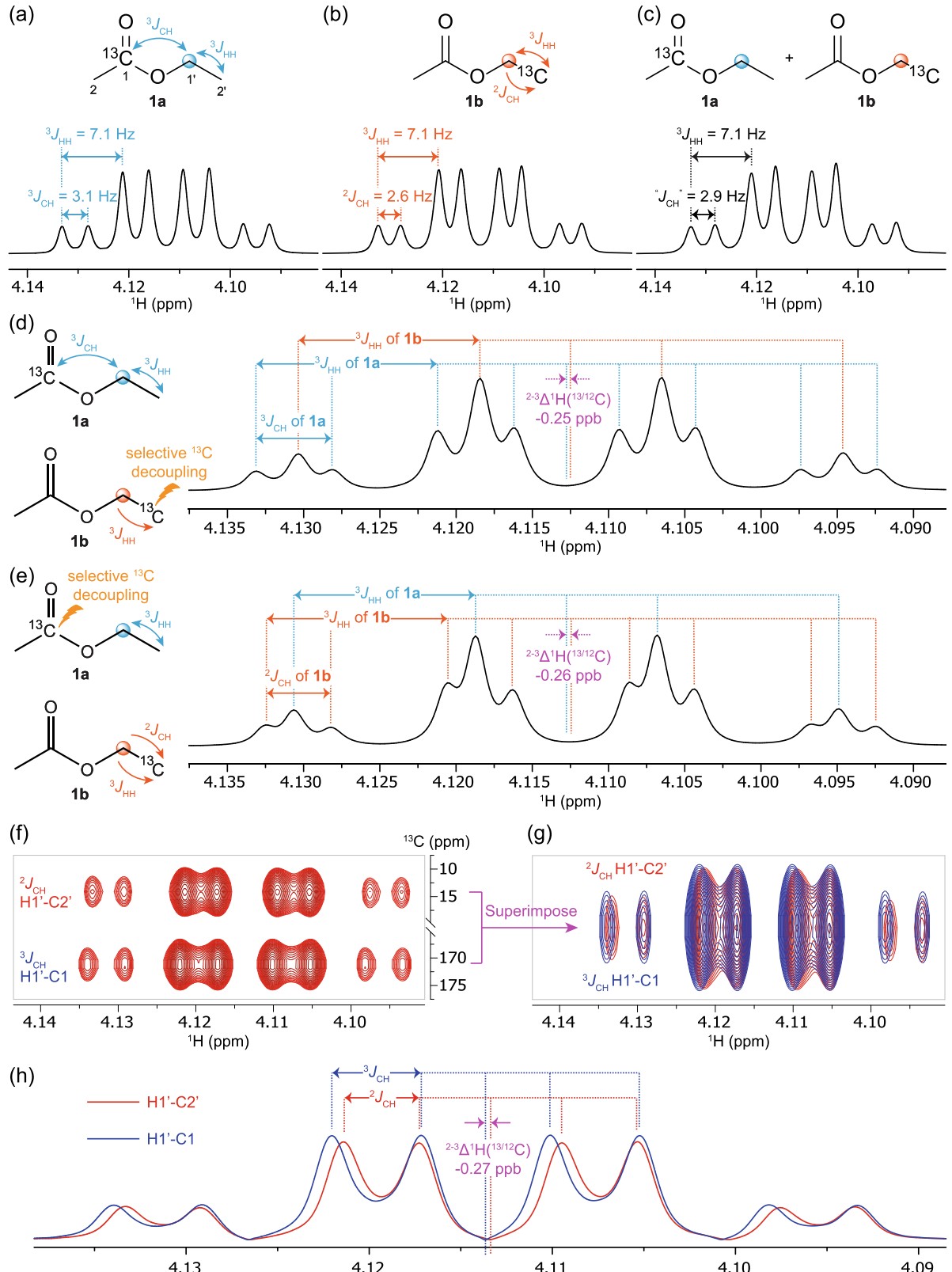

**Fig. 2 | Measurements of $^{2\text{-}3}\Delta^1H(^{13/12}C)$ of EtOAc.** Showing H1' of (**a**) **1a** (light blue), (**b**) **1b** (orange), (**c**) **1a** + **1b**, (**d**) **1a** + **1b** with C2' selective decoupling, (**e**) **1a** + **1b** with C1 selective decoupling (isotope shifts are magenta); (**f**) HMBC of EtOAc, zoomed in on H1'-C2' and H1'-C1 regions; (**g**) superimposition of H1'-C2' and H1'-C1 HMBC correlations; (**h**) overlaid HMBC horizontal slices of H1'-C2' and H1'-C1.

**Table 1 | $^{2\text{-}3}\Delta^1H(^{13/12}C)$ measurements of EtOAc at natural isotope abundance**

| Entry[a] | Pulse sequence | $^1H$ resonance frequency (MHz) | AQ (s) | $^{2\text{-}3}\Delta^1H(^{13/12}C)$ (ppb) |
|---|---|---|---|---|
| 1 | HMBC | 800 | 4.0 | −0.27 ± 0.01 |
| 2 | HMBC | 800 | 2.0 | −0.27 ± 0.01 |
| 3 | HMBC | 800 | 1.0 | −0.27 ± 0.01 |
| 4 | HMBC | 800 | 0.5 | −0.25 ± 0.01 |
| 5 | HMBC | 800 | 0.2 | −0.18 ± 0.06 |
| 6 | D-HMBC | 800 | 2.0 | −0.26 ± 0.01 |
| 7 | HMBC | 600 | 2.0 | −0.27 ± 0.01 |
| 8 | HMBC | 500 | 2.0 | −0.25 ± 0.01 |

[a]each experiment was run in triplicate.

widths overwhelmed the subtle differences in isotope shifts, and thus H1' still appeared as a quartet of doublets with averaged coupling constants (7.1 Hz and 2.9 Hz) without peak separation (Fig. 2c). To differentiate the two sets of signals of **1a** and **1b** in the mixture, low-power selective $^{13}C$ adiabatic decoupling of either C2' or C1 was employed using a decoupling power 46.12 dB lower than the 90° hard pulse power. This removed the doublets caused by $J_{CH}$ (Fig. 2d, e), such that H1' appeared as a superimposition of a qd and a quartet. In this way, the chemical shift difference of the qd and doublet in an identical environment can be accurately measured via line fitting. When C2' was decoupled, the $^{2\text{-}3}\Delta^1H(^{13/12}C)$ was measured to be −0.25 ppb with H1' of **1b** being more upfield; when C1 was decoupled, the $^{2\text{-}3}\Delta^1H(^{13/12}C)$ was measured to be −0.26 ppb. The consistent results of −0.25 to −0.26 ppb between the two measurements validated the accuracy of this method.

Next, we explored the possibility of utilizing the $^1H$-$^{13}C$ HMBC experiment to accurately measure $^{2\text{-}3}\Delta^1H(^{13}C/^{12}C)$ of EtOAc at natural isotopic abundance. Extending the acquisition time (AQ) of the free induction decay (FID) to boost the resolution of the F2 frequency domain after Fourier transformation could be an effective strategy for observing small isotope shifts less than 10 ppb[39–41]; therefore, an HMBC experiment with a 2.0 s AQ was attempted on EtOAc at natural isotopic abundance. By stacking the HMBC signals of H1'-C2' and H1'-C1, the peak position of H1'-C2' was shifted slightly upfield when compared to H1'-C1 (Fig. 2f, g). For a clearer visualization, the horizontal slices of both HMBC correlations were extracted and then superimposed allowing the peak shift to be easily observed (Fig. 2h). This approach also allowed more precise measurement of the chemical shift of each HMBC correlation from the multiplet peak positions. In this manner, the $^{2\text{-}3}\Delta^1H(^{13/12}C)$ was measured as −0.27 ppb, which was consistent with the value obtained from the $^{13}C$-labeled EtOAc, **1a** and **1b**. With zero-filling in the $^1H$ dimension to 64k complex points for a digital resolution of 0.15 ppb, the accuracy of $^{2\text{-}3}\Delta^1H(^{13/12}C)$ measurements via i-HMBC are reliably in the hundredths of a ppb provided that the signal-to-noise ratio is at least 30–35 (Table S12).

After establishing proof of concept for i-HMBC, which utilizes an extended acquisition time to detect $^n\Delta^1H(^{13}C/^{12}C)$, we next explored the impact of the AQ time, the effect of selective decoupling, and the NMR field strength on the i-HMBC methodology (Table 1). Each experiment was performed in triplicate to evaluate experimental reproducibility. First, AQ was decremented from 4.0 to 0.2 s (entries 1–5 in Table 1). An AQ of 1.0 s still produced satisfactory results. However, when the AQ was reduced to 0.5 s, acceptable isotope shift differences were still obtained but required perfectly symmetric line shapes for reproducibility. Shortening the AQ to 0.2 s caused the measured differences to become irreproducible and unreliable. Considering that HMBC spectra are typically acquired using 0.1–0.4 s AQ, suggests that it is almost impossible to observe such isotope shifts with the nominal parameter settings.

Next, we tried to remove the $J_{CH}$ splitting in the spectra via the decoupled-HMBC (D-HMBC) experiment using a slightly modified D-HMBC pulse sequence with an added low-pass J-filter and a CLIP element[43–45], in combination with mild adiabatic decoupling (entry 6, Table 1). With a 2.0 s AQ, the heating generated from decoupling did not adversely affect the data quality, and the qd patterns in all spectra were simplified to quartets with the same isotope shift differences. We also repeated the HMBC experiments using spectrometers ranging from 500 to 800 MHz (entries 2,7, and 8, Table 1). The consistent values showed that even a 500 MHz spectrometer can provide sufficient resolution to measure isotope shift differences accurately. Considering that spectrometers at higher field strengths or instruments with the newest versions of hardware might not be readily available, we chose to acquire i-HMBC data with an AQ of at least 1.0 s on a 600 MHz spectrometer equipped with a helium-cooled cryoprobe for the remainder of the examples reported.

Instead of the challenging measurement of the absolute value of the $^2\Delta H(^{13/12}C)$, we found that the difference between $^2\Delta^1H(^{13/12}C)$ and $^n\Delta^1H(^{13/12}C)$ ($n \geq 3$), $^{2\text{-}n}\Delta^1H(^{13/12}C)$, could be precisely determined by either i-HMBC or D-i-HMBC and used for identification of two-bond HMBC correlations. The terpene linalool (**2**) was used as a model compound to illustrate how this methodology works (Fig. 3). The methyl-H9 in **2** showed five HMBC correlations, including a two-bond correlation to C3, two three-bond correlations to C2 and C4, and two four-bond correlations to C1 and C5. With routinely employed HMBC parameters, the peak widths of the HMBC correlations were so wide (>7 Hz) that the $J_{CH}$ was not resolved due to low resolution on the direct (F2) dimension (Fig. 3a). In contrast, the high resolution provided by i-HMBC not only resolved three large $J_{CH}$ couplings, but the upfield isotope shift of H9-C3 was also clearly observed (Fig. 3b). Because the *precision* of determining peak positions using either direct peak picking or line fitting was as low as 0.01 ppb (Fig. 3d, e), corresponding isotope shift differences could be derived. The $^1H$ chemical shift of the HMBC correlations from H9 to C5, C4, C3, C1, and C2 were measured to be 1.296*7*0, 1.296*6*0, 1.296*0*8, 1.296*6*8 and 1.296*6*0 ppm, respectively (Fig. 3f). Among them, the H9-C5 correlation was the most downfield. Compared with this peak, the other four correlations showed −0.10, −0.62, −0.02, and −0.10 ppb isotope shift differences. The −0.62 ppb upfield shift of H9-C3 identified this as a two-bond correlation. D-i-HMBC simplified the spectra by removing the $J_{CH}$ splitting, transforming the five HMBC correlations into singlets (Fig. 3c). This simplified the peak picking/line fitting process and made the isotope shift difference of the two-bond correlation of H9-C3 more apparent.

To gain a better understanding of the magnitude of $^{2\text{-}n}\Delta^1H(^{13/12}C)$, i-HMBC spectra of a range of structurally diverse natural products and pharmaceutical compounds (**2-9**) were acquired (Fig. 4). For each proton, the HMBC correlation bearing the most downfield $^1H$ chemical shift was chosen as the reference, and isotope shift differences of other HMBC correlations from the same proton were measured as a relative value to the reference (Fig. 4a). For three-bond and longer-range HMBC correlations, the absolute values of the isotope shift differences compared to the references were usually found to be less than 0.3 ppb. In contrast, the absolute values of the isotope shift differences of two-bond HMBC correlations were generally above 0.3 ppb, and some were as large as 1.6 ppb. For easier identification of the two-bond HMBC correlations, the isotope shift differences (gap, as illustrated in the row of H1 of prednisone (**8**) in Fig. 4a) between two-bond and the closest longer-range correlations were summarized (Fig. 4b). In all cases the two-bond HMBC correlations were found to be at least 0.1 ppb more upfield than the corresponding three-bond and longer-range correlations from the same proton. In addition, 92% of the two-bond correlations were more than 0.2 ppb larger than the largest longer-range correlations. Based on these results, an isotope

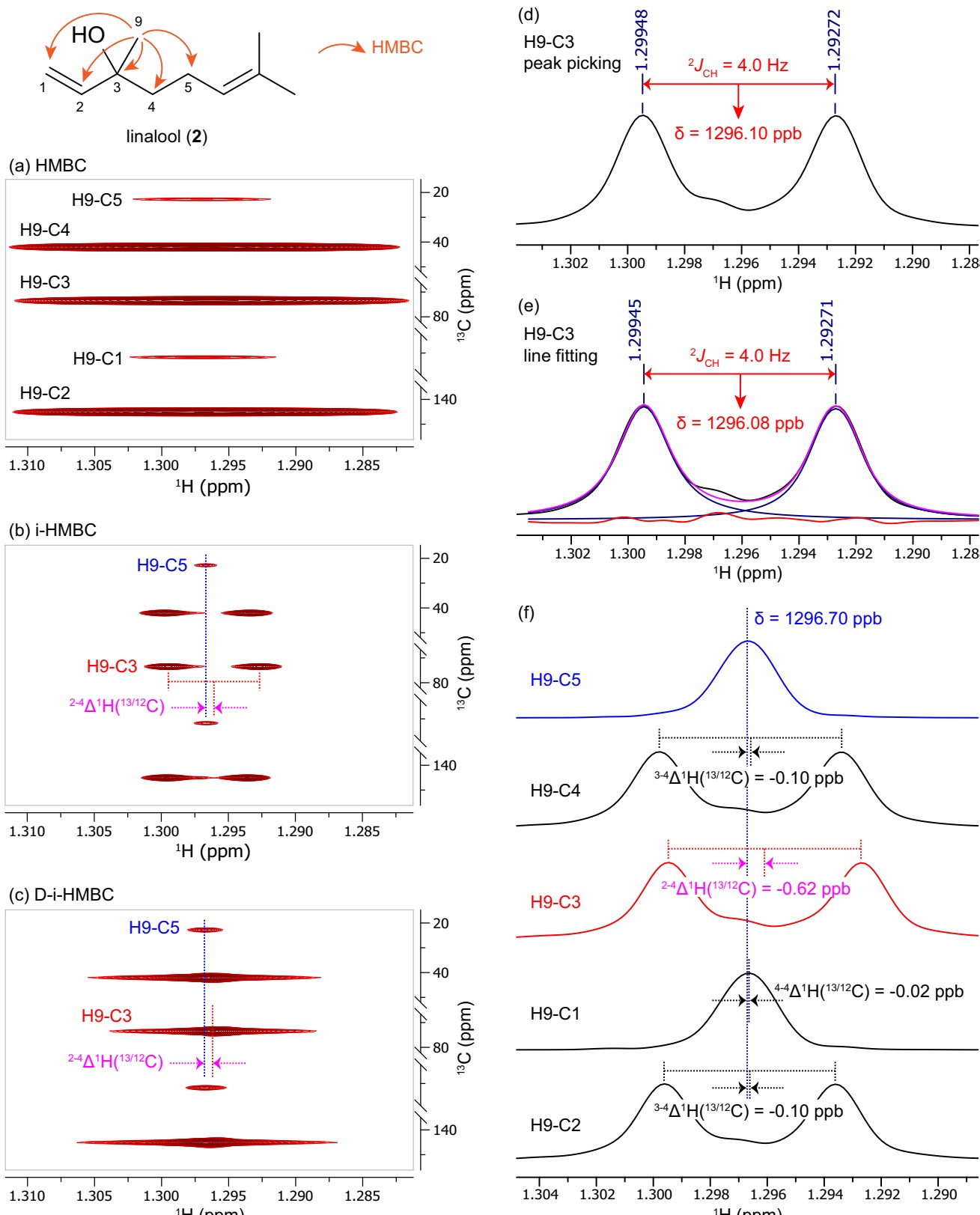

**Fig. 3 | Application of i-HMBC/D-i-HMBC methodology to identify the two-bond HMBC correlation of H9-C3 in linalool (2). a** nominal HMBC of H9; **b** i-HMBC of H9; **c** D-i-HMBC of H9 ($^2J_{CH}$ are in red, the most downfield $^4J_{CH}$ are in blue, the isotope shift difference are in magenta); methods to determine chemical shift (**d**) peak picking, (**e**) line fitting (blue: fitted peak; pink: sum of fitted peak; red: fitting residue); **f** stack of horizontal slices of i-HMBC correlations from H9, showing measurements of isotope shift differences.

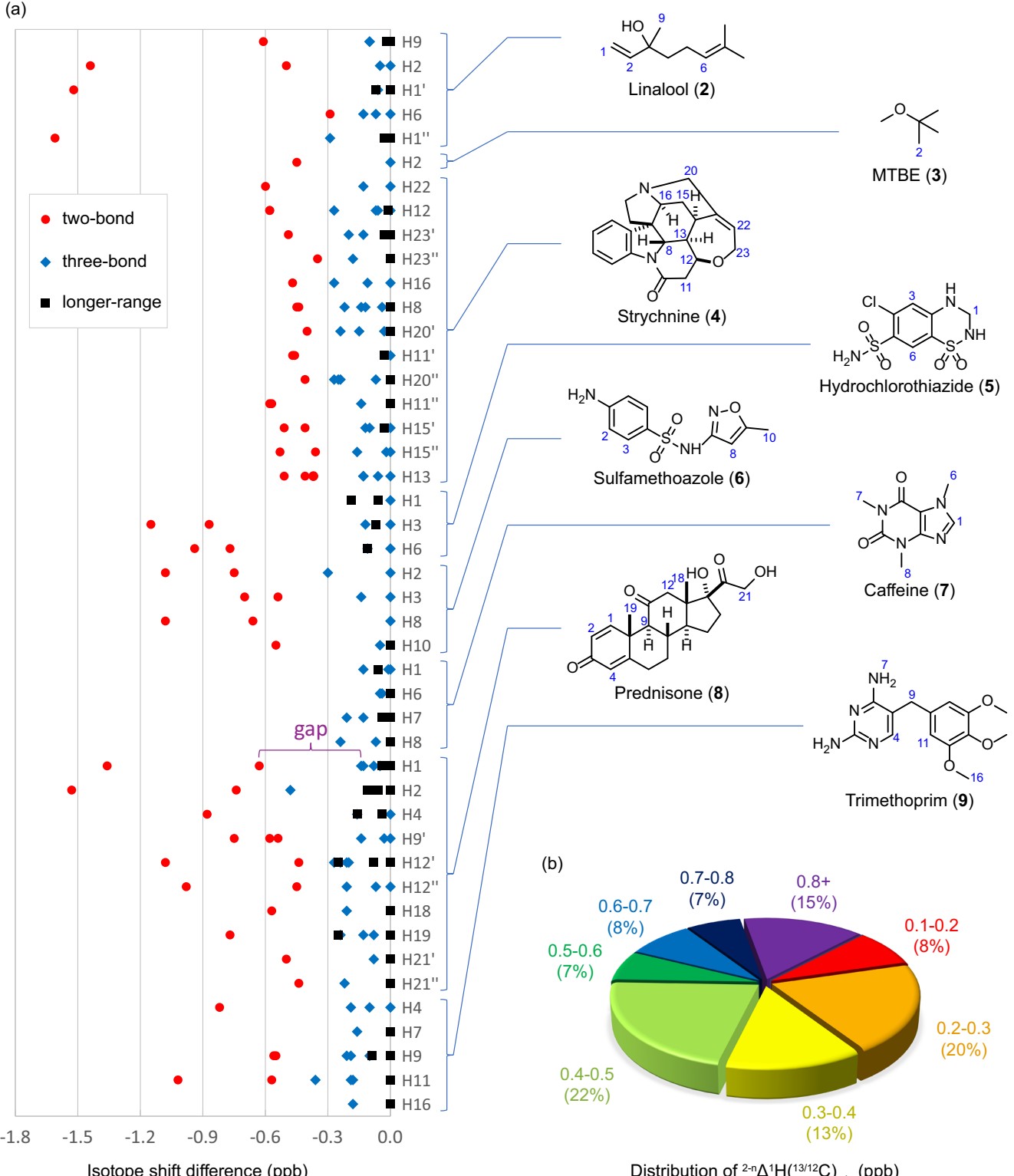

**Fig. 4 | Isotope shift differences observed in natural products and pharmaceuticals. a** Measured isotope shift difference in **2-9. b** Distribution of magnitudes of the isotope shift difference (gap) between two-bond correlation and the closest longer-range correlation ($^{2\text{-}n}\Delta^1H(^{13/12}C)_{min}$).

shift difference of ≥0.3 ppb from the reference and a 0.2 ppb gap from the closest signal can be used to efficiently and confidently identify two-bond HMBC correlations. With a reliable method to detect two-bond HMBC correlations in hand, applying i-HMBC to structure elucidation of structurally complex natural products was next examined.

**Application of i-HMBC to natural product structure elucidation**
Homodimericins are examples of severely proton-deficient molecules (H:C ratio <1) and are thus challenging for NMR structure elucidation (Fig. 5a). Homodimericin A (**10**) was first isolated from the fungus *Trichoderma harzianum*[46]. Its structure contains thirteen contiguous quaternary carbons, and the structure could not be elucidated with

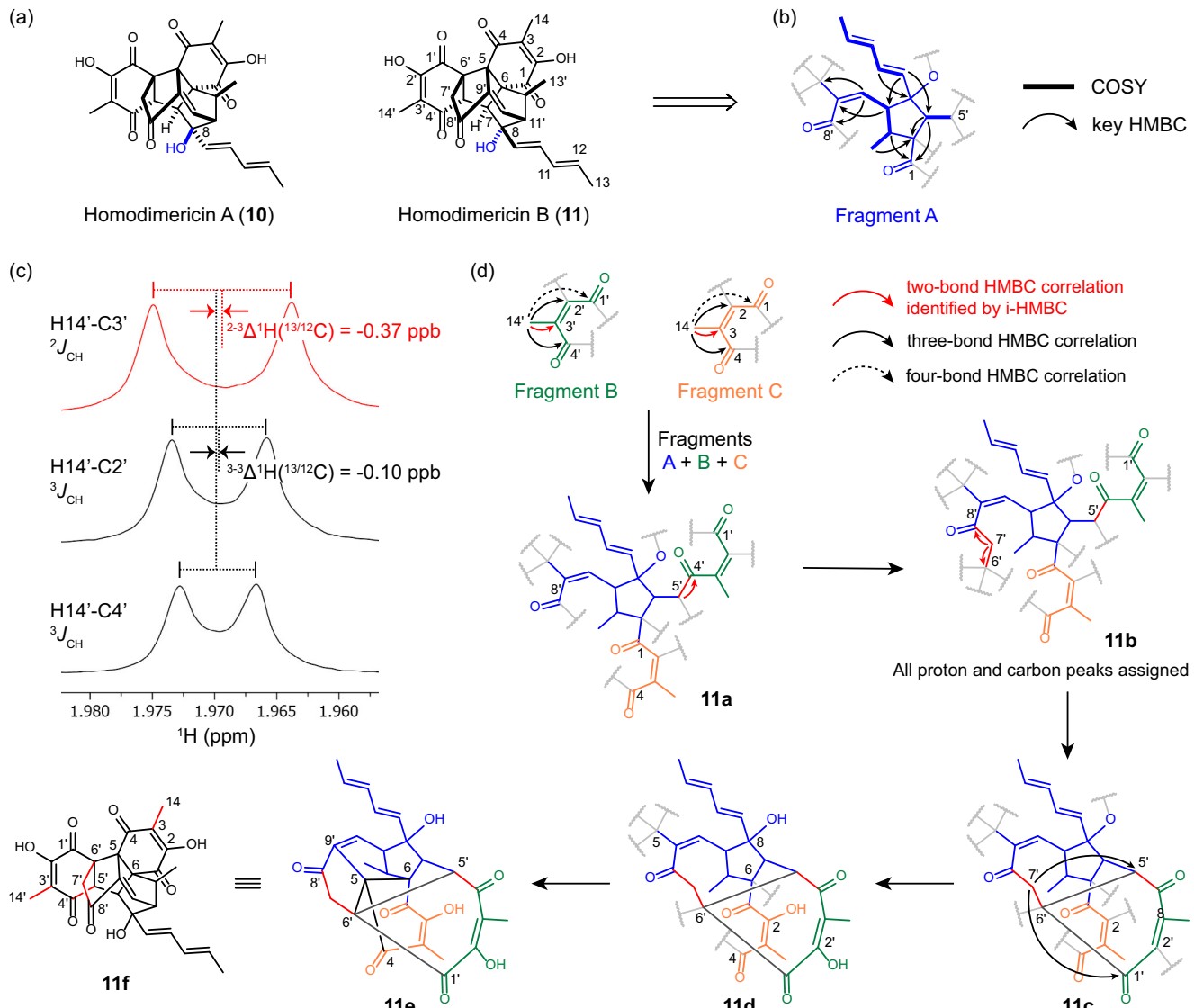

**Fig. 5 | Structure elucidation of homodimericin B (11) using i-HMBC. a** Structures of homodimericins A (**10**) and B (**11**) with stereoconfiguration difference indicated in blue; **b** fragment A was the only segment that could be solved using the conventional suite of 2D NMR experiments; **c** identification of the two-bond $^1$H-$^{13}$C correlation of H14'-C3'; $^2J_{CH}$ was shown in red; **d** structure elucidation of **11** using i-HMBC; key bond connectivities identified by i-HMBC were highlighted in red in **11 f**.

conventional 2D NMR techniques (*viz.*, COSY, HSQC, HMBC, and ROESY) or by using those data in conjunction with computer-assisted structure elucidation (CASE) algorithms. Two-bond proton-carbon correlations derived from the analysis of 1,1-HD-ADEQUATE[17] data were crucial; only after the two-bond proton-carbon correlations were incorporated into the data analysis could the structure of homodimericin A be fully elucidated. The structure of **10** was then orthogonally confirmed by anisotropic NMR[47], as well as total synthesis from four independent groups[48–51].

Homodimericin B (**11**), which was originally named 8-*epi*-homodimericin A, differs from **10** in the stereochemistry of the C8 hydroxy group, and was first identified as a side-product in the biomimetic total synthesis of **10**[50]. Its structure was proposed based on the synthetic route, HRMS and comparison of the NMR spectra with **10**. Homodimericin B (**11**) was then identified and isolated from the same fungus, *T. harzianum*, showing consistent NMR spectra with the synthetic sample[52]. Rigorous characterization of the complicated skeleton of **11** was still lacking, so we attempted to utilize i-HMBC to unambiguously confirm the structure. With only 0.47 mg (1.0 μmol as determined by PULCON[53] qNMR using 1,3,5-trimethoxybenzene as an external

standard) of **11**, we first started with conventional 2D NMR experiments. As expected, due to the highly proton-deficient nature of the compound, only fragment A (Fig. 5b) could be proposed based on the 2D NMR correlations.

i-HMBC data with a 2.0 s AQ were then acquired to facilitate the structure elucidation. In the i-HMBC spectra, Me14' exhibited three strong HMBC correlations with C3' (126.8 ppm), C2' (154.6 ppm) and C4' (194.9 ppm). The HMBC correlation between Me14' and C4' exhibited the most downfield shift in the $^1$H dimension and thus was used as the reference. Compared to this signal, the correlation between Me14' and C3' was shifted upfield by −0.37 ppb, while that of Me14' and C2' were only shifted upfield by −0.10 ppb (Fig. 5c). Consequently, the Me14'-C3' signal was identified as a two-bond HMBC correlation, which established that C3' is directly connected to C14'. The chemical shifts of C3', C2', and C4' indicated a highly polarized unsaturated ketone with a Me14' branch. Furthermore, a weak HMBC correlation between Me14' and C1' (192.9 ppm) suggested a four-bond HMBC correlation, and thus ketone C1' was connected to C2' to form fragment B (Fig. 5d). A structurally identical fragment, C, was observed with very similar chemical shifts to those of fragment B by using the same i-HMBC analysis methodology.

**(a)**

Calicheamicin γ₁ᴵ (**12**)

key two-bond i-HMBC (red) to elucidate
the proton-deficient core of **12**

**(b)**

Cryptospirolepine (**13a**)
original (1993)

Cryptospirolepine (**13b**)
revised (2015)

i-HMBC (red) confirmed the H13-C12a
HMBC is <u>not</u> a two-bond correlation
as proposed in the original
cryptospirolepine structure **13a**

**Fig. 6 | Applications of i-HMBC for structure confirmation of complex natural products. a** The proton-deficient core in calicheamicin γ₁ᴵ (**12**) and **b** the revised structure of cryptospirolepine (**13b**).

Fragments A and C share the same carbonyl, C1. A two-bond HMBC correlation from H5′ in fragment A to C4′ in fragment B, characterized by a −0.30 ppb isotope shift, established the connection between C4′ and C5′. Thus, the three fragments could be combined into a single fragment, **11a**. In addition, a methylene, CH₂-7′, exhibited a two-bond HMBC correlation to C8′, characterized by a −0.49 ppb isotope shift, which established the connection between C7′ and C8′. CH₂-7′ also showed a two-bond HMBC correlation with a −0.57 ppb isotope shift to a quaternary carbon C6′, yielding the intermediate structure **11b**. Considering that H7′ had strong HMBC correlations with C1′ and C5′, a network of C7′-C6′, C6′-C5′, and C6′-C1′ could be discerned as shown by **11c**. At this stage, all carbons and non-exchangeable protons had been assigned. The molecular formula of **11c** is $C_{28}H_{23}O_6$. Compared to the formula $C_{28}H_{26}O_8$ for **11**, it was obvious that the C8 position should be a tertiary alcohol. Two additional hydroxyl groups needed to be located. As mentioned above, the olefinic bonds between C2 and C3, and C2′ and C3′ were highly polarized, thus the two missing hydroxyl groups had to be placed at the open valences at C2 and C2′, affording structure **11d**. Finally, the three open valences of C5 had to connect with the open valences of C4, C6, and C6′, respectively, forming the complicated structure **11e**, which, after being redrawn, is recognizable as homodimericin B (**11 f**). The structure, **11 f**, was identical to the carbon skeleton of homodimericin A (**10**). A NOESY experiment (Supplementary Fig. 9) confirmed that the C8 position was epimerized compared with **10**, and thus established the proposed structure of homodimericin B (**11**).

A total of seven two-bond HMBC correlations were identified by i-HMBC, based on their 0.28–0.45 ppb upfield shifts relative to the longer-range HMBC correlations. Five of the seven correlations as shown in **11 f** (denoted by red bonds in the structure) were used and found to be crucial for structure elucidation of the proton-deficient core of **11**. When the two-bond correlations were identified by i-HMBC,

the previously formidable structure elucidation of **11** was dramatically simplified to a straightforward logical deduction process. Neither guesswork nor trial-and-error of atom linkages were necessary, both of which are commonly encountered in the structure elucidation of even much simpler compounds when confounded with multiple structural possibilities due to HMBC correlation path-length ambiguities. Compared to 1,1-ADEQUATE (or the homodecoupled variant), i-HMBC provides essentially the same structural information but is significantly more sensitive. Although the i-HMBC spectrum of **11** was acquired in 6 h, a 3 h acquisition was found to be sufficient to identify two-bond HMBC correlations. In contrast, after a 90 h acquisition of 1,1-ADEQUATE data using the same sample and NMR spectrometer, only 9 weak correlations were observed with a S/N of 2.5–3.5 for methylenes and methines and a S/N of 3.0–6.5 for methyl groups. Among them, five were from the proton-rich region, where the information could readily be obtained by COSY. Only four correlations were related to the proton-deficient core. Moreover, in the seminal work of the structure elucidation of homodimericin A (**10**), a 41 h data acquisition using a 6 mg (13 μmol) sample in a 1.7 mm MicroCryoProbe® was necessary to obtain a good quality 1,1-HD-ADEQUATE spectrum[46]. Thus, i-HMBC is demonstrably more capable of providing comparably useful carbon-carbon bond connection information when compared with 1,1-ADEQUATE while requiring only 7% of the acquisition time.

To further demonstrate the generality of this method, the highly complex structures of enediyne calicheamicin γ₁ᴵ (**12**, 5 mg, 3.7 μmol)[54–58] and the spiro-nonacyclic indole alkaloid cryptospirolepine (**13b**, 0.14 mg, 0.28 μmol)[17,59] were also confirmed by i-HMBC (Fig. 6, Supplementary Figs. 16 and 18). These natural products were not able to be structurally characterized by the conventional suite of 2D NMR experiments. Moreover, the original structure of cryptospirolepine was incorrect due to misidentification of a two-bond HMBC correlation[17].

In this work, we have shown i-HMBC methodology can differentiate two-bond from three-bond and longer-range HMBC correlations by reliably and accurately measuring sub 0.1 ppb isotope shifts and the corresponding isotope shift differences, on nanomole quantities of complex natural products. Unequivocal identification of two-bond HMBC correlations is crucial for characterization of highly complex proton-deficient structures because they provide key information of carbon-carbon bond connection. We demonstrated this methodology to be highly powerful and practical through structure confirmation of homodimericin B (**11**), calicheamicin $\gamma_1^I$ (**12**), and the revised structure of cryptospirolepine (**13b**). Compared with 1,1- and 1,1-HD-ADEQUATE data, which provides very similar information, i-HMBC is much more sensitive and could be used for sub-mg/nanomole materials with reasonable data acquisition times of several hours (Table S11). Considering that the recycle time for the nominal HMBC experiment is typically 1.5–2.5 s (0.2–0.4 s of AQ and 1.0–2.0 s for the relaxation delay, D1), i-HMBC is only slightly longer (1.0–2.0 s of AQ and 0.5 s of D1). Thus, we propose the use of i-HMBC as a complement to HMBC for NMR structure elucidation because it provides important isotope shift information at minimal extra cost of total experiment time (also note that if a decreased S/N caused by relaxation was a concern, then the FID could be simply truncated during spectral processing to generate a nominal HMBC spectrum). In addition, methodology for measuring relative $^{1-n}\Delta^1H(^{13/12}C)$ and $^{2-n}\Delta^1H(^{13/12}C)$ isotope shifts developed in this work could also be utilized with other heteronuclear NMR experiments, such as HSQMBC, HMQC (D-i-HMBC) and ADEQUATE[60]. Broader adoption of this methodology should lead to more published $\Delta^1H(^{13/12}C)$ isotope shift data, which, in turn, will undoubtedly enhance the body of knowledge for this fundamental yet thus far underexplored measurable NMR parameter.

## Methods

NMR spectra were acquired on the following instruments: a Bruker 800 MHz Neo NMR spectrometer equipped with a 5-mm TCI CryoProbe™, a Bruker 600 MHz AVIII HD NMR spectrometer equipped with a 5-mm TCI CryoProbe™, a Bruker 600 MHz Neo NMR spectrometer equipped with a 5-mm QCI-P CryoProbe™, and a Bruker 500 MHz AVIII HD NMR spectrometer equipped with a 5-mm Prodigy™ Probe.

The typical procedure for i-HMBC acquisition and processing was as follows. A phase sensitive echo/antiecho HMBC pulse sequence (hmbcetgpnd) was used with following parameters: for $^1H$ dimension (at 600 MHz), 10 ppm spectrum width (SW), 24036 points (TD) or 12018 complex points, 2.0 s AQ, resulted in 0.5 Hz digital resolution (FIDRES); for $^{13}C$ dimension, 220 ppm SW, sufficient points to resolve carbon signals (512 increments and 16 number of scans were used for homodimericin B (**11**)); 0.5 s relaxation delay (D1). When S/N decrease caused by T2 relaxation is a concern as in calicheamicin (**12**) and cryptospirolepine (**13b**), 1.0 s AQ (1.0 Hz FIDRES) was used. During processing, a 90° sine square apodization was applied in the F1 dimension, and a 45° sine bell was applied in F2; 4x zero-filling was used to increase the number of points in the $^1H$ dimension to 64k complex points (8x zero-filling was used for 1.0 s AQ) for a digital resolution of 0.09 Hz (0.15 ppb). The exact i-HMBC peak positions (in ppm, with 5 decimals) were obtained by extracting the horizontal slices followed by peak picking or line fitting using MestreLab MNova software (version 14.2) (also see SI section IX). A S/N > 30–40 was generally necessary for accurate measurement of isotope shift differences. If peak shape distortion is too significant due to the lock being disturbed by the application of gradient pulses, a slightly modified HMBC pulse sequence in combination with a lower lock power might help to improve the peak shape (see SI section VII).

## Data availability

The complete NMR spectra, acquisition and processing parameters, pulse sequences, and additional experimental details are provided in the Supplementary Information file.

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

## Acknowledgements

The authors thank Drs. Alexei V. Buevich (Merck & Co., Inc) and Zhang Wang (Arcus Biosciences) for helpful discussions. The authors would like to thank the MRL Analytical Research and Development department for making available the use of instrumentation in this study.

## Author contributions

X.W. conceived the methodology. X.W., Y.W., and R.D.C. performed the NMR experiments. X.W. and G.D.P. coded pulse sequences. Z.H. and H.Y. synthesized the $^{13}$C-labeled EtOAc. A.F. isolated and purified homodimericin B. The manuscript was written by X.W., R.D.C., Y.W., M.R., and G.E.M. The project was supervised by X.W., M.R., and E.C.S.

## Competing interests

The authors declare no competing interests.
