## [Peer Review File · Nature Communications]

REVIEWER COMMENTS

Reviewer #1 (Remarks to the Author):

This article describes the use of isotopic ^{13}C shift effects in ^1H NMR to solve an important problem in structural elucidation by NMR consisting in differentiating two-bond heteronuclear correlations with respect to three-bond connectivities in HMBC spectra. Although the proposed methodology is based on a simple setting change (long AQ to achieve ultra high levels of digital resolution in a conventional 2D HMBC spectrum) and a known experimental observation that was already proposed years ago (see reference 36), the proposed strategy can be considered brilliant and will allow to have a new NMR protocol to solve a very relevant structural problem in the structural elucidation of complex molecules. The article is well written and structured, with very illustrative examples, which could be accepted for publishing in Nature Communications after considering some important aspects. There are some questions that need to be clarified to avoid confusion for non-specialized NMR readers because there is no a rigorous discussion on several very important aspects related to the accurate determination of such very small isotopic chemical shift differences

1. The authors say that iHMBC should replace the conventional and widely accepted way of using the HMBC experiment in structural elucidation studies. I think that this statement, which is found in the abstract and in the conclusions, is not appropriated and should be removed. iHMBC cannot be presented as a substitute for routine HMBC, but as a complement to help when it is necessary to differentiate 2J vs 3J. There are several reasons for this: lower sensitivity, longer experimental times, very large data storage and debatable use of NUS.

2. Resolution and sensitivity have opposite sense in NMR. The sensitivity of the iHMBC experiment is not discussed in a rigorous way, and it is wrongly commented that it is similar to the HMBC. Authors should focus on the term SNR per time unit and it can be predicted that iHMBC suffers from a very strong penalty in sensitivity with respect to conventional HMBC. Authors must be required to provide experimental data to quantify the real loss of sensitivity of the iHMBC experiment with respect to a classic experiment acquired at the same experimental time. It should be interesting to compare the experimental SNR of two experiments acquired with the same experimental time and with the same scans: the classic obtained with $\text{TD}=2048$, $\text{D1}=2\text{s}$, $\text{AQ}=0.2\text{s}$; and the iHMBC with $\text{TD}=20480$, $\text{D1}=0.2$ and $\text{AQ}=2\text{s}$. If what I think happens, increasing the AQ value by an order of magnitude would result in sensitivity losses $>90\%$. To achieve an iHMBC with enough quality, a major number of scans (longer experimental time) would be required compared to a routine HMBC. If not, the low sensitivity of the iHMBC can make it difficult to observe tiny SNR cross-peaks with small $n\text{JCH}$ values (especially 2JCH) due to signal cancellation by mixed $i\text{P}/\text{AP}$ contributions and due to its sine intensity dependence.

3. In the experimental part there is no clear a description of the basic acquisition parameters. What does 24036 points mean? are they real/complex? Nor is there a clear description of the very important digital resolution in the time dimension (FIDRES) parameter, which is what really marks the degree of accuracy of the measure.

4. There is no comments on the accuracy and the measurement error, and this is a very important factor because the authors propose to measure very small deviations of only a few Hz. In a 500 Mz spectrometer, a FIDRES of 0.1 ppb (AQ=2s) represents 0.5 Hz and therefore there is a detection limit that should be also considered. On the other hand, an experiment with AQ=1s represents a FIDRES of 1Hz (0.2ppb) which, in view of what is intended to be measured, represents a very high percentage of inaccuracy or error. I understand that this option does not seem optimal.

5. The experimental part comments on the use of high levels of zero-filling, which increases HzPt in the final spectrum, but which at a practical level have a highly cosmetic aspect. The value to consider is FIDRES and I'm not convinced that increasing the zero filling x4 to achieve a fictitious spectral resolution of 0.01ppb offers a better measure. Can we measure 0.02ppb when FIDRES is 0.1ppb?

6. In the figures, it is not mentioned that the 1D slices are represented in magnitude mode. Nothing is said about the mixed phases of HMBC cross-peaks, resulting of a mixture of IP and AP contributions. In high resolution conditions as found in iHMBC, this becomes much more critical, especially in analysis and fitting tasks of very complex and, also very important, asymmetric multiplet structures that blur the exact position of the center of the multiplet. I think that an accurate determination of the shift would only be possible using a second reference spectrum.

7. I think it is not recommended to use NUS in iHMBC to reduce the experimental time. The attempt to analyze and determine small deviations of some Hz with certain precision in highly complex cross-peaks resulting from a reconstructed spectrum is highly dangerous, especially when the SNR is not high, which easily can induce the introduction of signal distortions and a source of significant error in the measurement.

Reviewer #2 (Remarks to the Author):

The manuscript demonstrates that by recording HMBC spectra at very high resolution in the ^1H (detected) dimension, it is possible to distinguish 2- from 3-bond (and longer range) correlations. The finding that the two-bond isotope shift on ^1H is actually measurable and consistently larger than the minute 3-bond isotope shift follows expectation, but the fact that it is readily extracted from the widely used HMBC spectrum provided that it is recorded with sufficient ^1H resolution is highly remarkable and useful. The fact that the HMBC spectrum provides internal referencing makes the method quite robust.

A logical follow-up to this work (future work) would be to compare the reported isotope shifts (SI) to the structural features, such that it becomes possible to predict the expected magnitude of the two-bond isotope shift on the basis of structure and thus validate conclusions.

The authors demonstrate the method for an impressive number of highly practical examples, the manuscript is well written, and I expect this innovation will become widely used.

Reviewer #3 (Remarks to the Author):

The manuscript describes measurement of isotope effects in the well-known NMR experiment HMBC and uses the relative size of isotope effects to distinguish two-bond correlations from correlations with more than two intervening bonds. This use of isotope effects is new and correct identification of the number of bonds between atoms being correlated is essential in structure elucidation work.

Let me put up and discuss two questions for judging possible publication of this manuscript in Nature Communications:

1) Is the method proposed going to be the preferred approach for identification of two-bond correlations?

The authors mention the alternative 1,1-ADEQUATE which is totally safe, easy and reliable but requires two ^{13}C nuclei per molecule. However, this does not mean a sensitivity disadvantage of two orders of magnitude because the HMBC sensitivity is strongly dependent on the size of $nJ(\text{CH})$. In addition, the authors state that their method requires a spectral signal-to-noise ratio in excess of 30 which is much more than is needed with 1,1-ADEQUATE. In the limit of a vanishing $2J(\text{CH})$ the proposed method will fail, but 1,1-ADEQUATE will work.

Figure 3a shows that there is overlap between two- and three-bond isotope shift differences making the distinction a priori ambiguous, something the authors argue around in lines 204-209, but there is no proof or evidence that this solution is generally applicable.

All in all I think routine NMR users will hang on to 1,1-ADEQUATE as the preferred method and live with a longer measurement time. It is so much easier to observe a peak or no peak at a given position than the more elaborate procedure of meticulously mapping out all the small isotope effects.

2) Will the broad readership and scope of this journal appreciate the work and its presentation?

The manuscript contains many technical details and descriptions about preliminary validation experiments, something the majority of the readers probably will not be interested in.

In conclusion, I find the research interesting and valid but recommend publication in a journal with a larger fraction of NMR spectroscopist readers.

Minor points:

- Inclusion of the values of the $nJ(\text{CH})$ coupling constants involved might be helpful and correlate with the isotope effects.

- How do the authors deal with situations without meaningful isotope shift differences, like when a proton shows only one correlation or the case with e.g. two two-bond correlations and nothing else? Measurement of individual isotope effects is referred to as challenging in line 168.
- In fragments with long chains of quaternary carbons only the INADEQUATE experiment will be useful, something that deserves to be mentioned.
- Have the authors considered other experiments than HMBC for measurement of the isotope effects of interest?

Recommendations to the authors:

- Work on putting the handling of the overlap between two- and three-bond isotope effects on more solid ground. It is a good start that your procedure seems to correctly predict the number of intervening bonds on the molecules studied but more is needed to claim general applicability of the method.
- Work out how you will deal with the situations where there are no meaningful isotope shift differences but only individual isotope shifts.
- Discuss the possible use of other experiments than HMBC for measurement of these isotope effects.

REVIEWER COMMENTS

Reviewer #1 (Remarks to the Author):

This article describes the use of isotopic ^{13}C shift effects in ^1H NMR to solve an important problem in structural elucidation by NMR consisting in differentiating two-bond heteronuclear correlations with respect to three-bond connectivities in HMBC spectra. Although the proposed methodology is based on a simple setting change (long AQ to achieve ultra high levels of digital resolution in a conventional 2D HMBC spectrum) and a known experimental observation that was already proposed years ago (see reference 36), the proposed strategy can be considered brilliant and will allow to have a new NMR protocol to solve a very relevant structural problem in the structural elucidation of complex molecules. The article is well written and structured, with very illustrative examples, which could be accepted for publishing in Nature Communications after considering some important aspects. There are some questions that need to be clarified to avoid confusion for non-specialized NMR readers because there is no a rigorous discussion on several very important aspects related to the accurate determination of such very small isotopic chemical shift differences

1. The authors say that iHMBC should replace the conventional and widely accepted way of using the HMBC experiment in structural elucidation studies. I think that this statement, which is found in the abstract and in the conclusions, is not appropriated and should be removed. iHMBC cannot be presented as a substitute for routine HMBC, but as a complement to help when it is necessary to differentiate 2J vs 3J. There are several reasons for this: lower sensitivity, longer experimental times, very large data storage and debatable use of NUS.

We thank the reviewer for this suggestion. We have adjusted the wording of the abstract and of the conclusions to address this comment. In addition, please refer to our responses to questions 4, 5 and 7 for a more detailed discussion about the S/N and use of NUS.

2. Resolution and sensitivity have opposite sense in NMR. The sensitivity of the iHMBC experiment is not discussed in a rigorous way, and it is wrongly commented that it is similar to the HMBC. Authors should focus on the term SNR per time unit and it can be predicted that iHMBC suffers from a very strong penalty in sensitivity with respect to conventional HMBC. Authors must be required to provide experimental data to quantify the real loss of sensitivity of the iHMBC experiment with respect to a classic experiment acquired at the same experimental time. It should be interesting to compare the experimental SNR of two experiments acquired with the same experimental time and with the same scans: the classic obtained with TD=2048, D1=2s, AQ=0.2s; and the iHMBC with TD=20480, D1=0.2 and AQ=2s. If what I think happens, increasing the AQ value by an order of magnitude would result in sensitivity losses >90%. To achieve an iHMBC with enough quality, a major number of scans (longer experimental time) would be required compared to a routine HMBC. If not, the low sensitivity of the iHMBC can make it difficult to observe tiny SNR cross-peaks with small $n\text{JCH}$ values (especially 2JCH) due to signal cancellation by mixed iP/AP contributions and due to its sine intensity dependence.

We thank the reviewer for this comment. Indeed, S/N is an important consideration, especially with a long acquisition time (AQ). We added quantitative comparison of S/N between HMBC, i-HMBC and 1,1-ADEQUATE to the SI (section VIII and Table S11). Our data demonstrates that for strychnine, used as a representative small molecule, i-HMBC with 2.0 s AQ achieved ~40-50% of the sensitivity in the same

total experiment time compared to the nominal HMBC experiment with 0.2 s AQ. We believe that a moderate S/N loss is an acceptable trade-off as we can still generate high quality i-HMBC spectra with sub-mg amount of material in a reasonable experiment time.

For compounds with fast T2 relaxation, we did use a shorter AQ in i-HMBC to achieve higher S/N. For both calicheamicin (MW = 1368 Da) and cryptospirolepine (relatively broad peaks), we used 1.0 s AQ and are still able to obtain usable isotope shifts. We have added the i-HMBC experimental parameters for these two compounds to the SI.

When comparing i-HMBC (2s AQ) versus 1,1-ADEQUATE, the current best method for identifying two-bond C-H correlations, i-HMBC is about 40x more sensitive even when $^2J_{CH}$ is as small as 2.2 Hz. The difference is significant enough to justify the assertion of the superiority of i-HMBC over 1,1-ADEQUATE. For a more detailed discussion, please see our response to the first comment of Reviewer #3.

3. In the experimental part there is no clear a description of the basic acquisition parameters. What does 24036 points mean? are they real/complex? Nor is there a clear description of the very important digital resolution in the time dimension (FIDRES) parameter, which is what really marks the degree of accuracy of the measure.

We thank the reviewer for this feedback. We clarified the description of the experimental acquisition parameters in the manuscript.

FIDRES is 1/AQ. The number of points describe the parameter TD, which is comprised of half real and half imaginary numbers. For TD=24036 points, there are 12018 real/imaginary or 12018 complex points. We have revised the methods and SI sections to be more accurate.

4. There is no comments on the accuracy and the measurement error, and this is a very important factor because the authors propose to measure very small deviations of only a few Hz. In a 500 Mz spectrometer, a FIDRES of 0.1 ppb (AQ=2s) represents 0.5 Hz and therefore there is a detection limit that should be also considered. On the other hand, an experiment with AQ=1s represents a FIDRES of 1Hz (0.2ppb) which, in view of what is intended to be measured, represents a very high percentage of inaccuracy or error. I understand that this option does not seem optimal.

We thank the reviewer for this insightful comment. We agree that accuracy of the measurement is a very important factor since measured values are in the sub-ppb range. We have expanded the discussion of the accuracy in the manuscript and added Section IX to the SI to address this. Table 1 in the revised manuscript and Table S12 in the SI show the accuracy and reproducibility of the $^{2-3}\Delta^1\text{H}(^{13/12}\text{C})$ measurements of EtOAc using i-HMBC. Since the same $^{2-3}\Delta^1\text{H}(^{13/12}\text{C})$ isotope shift was measured directly by 1D ^1H NMR using ^{13}C -labeled EtOAc, we can compare the values to ensure that i-HMBC measurements are reliable, reproducible, and accurate. As long as the S/N of the peak of interest is at least 30, the accuracy of i-HMBC-based measurements is in the hundredths of a ppb on a 600 MHz spectrometer. Details of spectral processing (zero-filling, resolution, line fitting) are discussed in our response to the next comment.

5. The experimental part comments on the use of high levels of zero-filling, which increases HzPt in the final spectrum, but which at a practical level have a highly cosmetic aspect. The value to consider is FIDRES and I'm not convinced that increasing the zero filling x4 to achieve a fictitious spectral resolution of 0.01ppb offers a better measure. Can we measure 0.02ppb when FIDRES is 0.1ppb?

We thank the reviewer for this comment as it provides an opportunity to clarify the measurement procedure. The manuscript and the SI have been updated to add the details of spectral processing. Zero filling (ZF) is a commonly used NMR spectral processing procedure to increase apparent digital resolution. Conventional processing includes doubling the apparent resolution by 1x ZF (after adding zeros to round the number of points to the next power of 2 necessary for the Fast Fourier Transform algorithm), and we also explored 2x ZF and 4x ZF. While we understand reviewer's skepticism regarding the resultant *apparent resolution* increase, the latter is quite helpful for the methodology that relies on accurate peak positioning. Since we are using line fitting (either explicit Lorentzian deconvolution or GSD peak picking in Mestrenova), the most important parameter is sufficient S/N (see above). Once S/N is above 30, the $^{2-3}\Delta^1\text{H}(^{13}/^{12}\text{C})$ value is measured with sufficient accuracy for 1x, 2x and 4x ZF (see Table S12). However, 4x ZF (zero filling to 64k complex points for a digital resolution of 0.09 Hz or 0.15 ppb at 600 MHz) is better for the measurement reliability and thus this is our recommended processing procedure.

6. In the figures, it is not mentioned that the 1D slices are represented in magnitude mode. Nothing is said about the mixed phases of HMBC cross-peaks, resulting of a mixture of IP and AP contributions. In high resolution conditions as found in iHMBC, this becomes much more critical, especially in analysis and fitting tasks of very complex and, also very important, asymmetric multiplet structures that blur the exact position of the center of the multiplet. I think that an accurate determination of the shift would only be possible using a second reference spectrum.

We thank the reviewer for providing an opportunity to clarify this point. All HMBC data are processed in magnitude mode, which is the conventional use of HMBC data for structure elucidation. The measured value is a *relative* isotope shift, which means the exact lineshape is not very important as relative measurement overcomes this issue. Figure 2b clearly demonstrates that a relative isotope shift can be observed and measured directly. Please see the accuracy discussion in the responses above.

7. I think it is not recommended to use NUS in iHMBC to reduce the experimental time. The attempt to analyze and determine small deviations of some Hz with certain precision in highly complex cross-peaks resulting from a reconstructed spectrum is highly dangerous, especially when the SNR is not high, which easily can induce the introduction of signal distortions and a source of significant error in the measurement.

We thank the reviewer for this insightful comment. We have included 50% NUS i-HMBC data as a part of our analysis of the measurement accuracy in Section IX of the SI. The results corroborate reviewer's concern: indeed, NUS is detrimental for the measurement's accuracy when S/N is not sufficiently high. The manuscript and the SI have been revised to address this point.

Reviewer #2 (Remarks to the Author):

The manuscript demonstrates that by recording HMBC spectra at very high resolution in the ^1H (detected) dimension, it is possible to distinguish 2- from 3-bond (and longer range) correlations. The finding that the two-bond isotope shift on ^1H is actually measurable and consistently larger than the minute 3-bond isotope shift follows expectation, but the fact that it is readily extracted from the widely used HMBC spectrum provided that it is recorded with sufficient ^1H resolution is highly remarkable and useful. The fact that the HMBC spectrum provides internal referencing makes the method quite robust. A logical follow-up to this work (future work) would be to compare the reported isotope shifts (SI) to the structural features, such that it becomes possible to predict the expected magnitude of the two-bond isotope shift on the basis of structure and thus validate conclusions.

The authors demonstrate the method for an impressive number of highly practical examples, the manuscript is well written, and I expect this innovation will become widely used.

Thank you for kind comments and feedback, particularly regarding suggested follow-up work. Indeed, we are currently working on predicting isotope shifts by quantum chemical methods and plan to compare them with experimentally measured data.

Reviewer #3 (Remarks to the Author):

The manuscript describes measurement of isotope effects in the well-known NMR experiment HMBC and uses the relative size of isotope effects to distinguish two-bond correlations from correlations with more than two intervening bonds. This use of isotope effects is new and correct identification of the number of bonds between atoms being correlated is essential in structure elucidation work.

Let me put up and discuss two questions for judging possible publication of this manuscript in Nature Communications:

1) Is the method proposed going to be the preferred approach for identification of two-bond correlations?

The authors mention the alternative 1,1-ADEQUATE which is totally safe, easy and reliable but requires two ^{13}C nuclei per molecule. However, this does not mean a sensitivity disadvantage of two orders of magnitude because the HMBC sensitivity is strongly dependent on the size of $nJ(\text{CH})$. In addition, the authors state that their method requires a spectral signal-to-noise ratio in excess of 30 which is much more than is needed with 1,1-ADEQUATE. In the limit of a vanishing $2J(\text{CH})$ the proposed method will fail, but 1,1-ADEQUATE will work.

Figure 3a shows that there is overlap between two- and three-bond isotope shift differences making the distinction a priori ambiguous, something the authors argue around in lines 204-209, but there is no proof or evidence that this solution is generally applicable.

All in all I think routine NMR users will hang on to 1,1-ADEQUATE as the preferred method and live with a longer measurement time. It is so much easier to observe a peak or no peak at a given position than the more elaborate procedure of meticulously mapping out all the small isotope effects.

We do recommend i-HMBC as the preferred approach for identification of two-bond correlations due to its much higher sensitivity compared to 1,1-ADEQUATE. We added S/N comparison of regular HMBC, i-HMBC and 1,1-ADEQUATE in Table S11. Our results show that achieving S/N > 30 for i-HMBC is in fact much easier experimentally than detecting any signals in 1,1-ADEQUATE due to ~40x sensitivity difference even when $^2J_{CH}$ is as small as 2.2 Hz. For sub-mg amount of material, which is often the case for novel natural products, 1,1-ADEQUATE is just not feasible even with state-of-art cryoprobes. Cryptospirolepine is the only natural product to-date that was elucidated by 1,1-ADEQUATE at sub-mg scale (0.7 mg) but required a shocking 120-hour data acquisition! In contrast, we demonstrated that i-HMBC is applicable to 470 micrograms homodimericin B with just 6 hours of data acquisition and 140 micrograms of cryptospirolepine with 57 hours of data acquisition. The same 140 microgram sample of cryptospirolepine would have required ~3000 hours (or 125 days) of data acquisition for 1,1-ADEQUATE experiment, which is, of course, entirely unreasonable.

However, we do not propose to completely replace 1,1-ADEQUATE by i-HMBC, due to the *J*-modulation issue mentioned by the reviewer. In fact, we showed in Scheme S2 that the two-bond i-HMBC correlation H13-C2 was not observed in cryptospirolepine due to the 0.03 Hz $^2J_{CH}$ (calculated by DFT). In this case, although i-HMBC could be used to disprove the originally proposed structure of cryptospirolepine, 1,1-ADEQUATE was still required to elucidate its real structure.

It should also be noted that 1,1-ADEQUATE is not a routine experiment with straightforward data interpretation as this experiment does not irrefutably provide *only* $^1J_{CC}$ correlations. There are several examples reported [e.g., *Chem Open*, **4**, 577-581 (2015), *J. Org. Chem.*, **80**, 7396-7402, (2015), *Magn. Reson. Chem.*, **54**, 341-345 (2016), *Magn. Reson. Chem.*, **54**, 897-900 (2016), *Magn. Reson. Chem.*, **56**, 775-781 (2018)] in which $^nJ_{CC}$ correlations are observed in typically optimized 1,1-ADEQUATE experiments. This is due to the fact that some $^nJ_{CC}$ coupling constants are large enough to give appreciable response intensity in 1,1-ADEQUATE. Such $^nJ_{CC}$ correlations could be easily misinterpreted as $^1J_{CC}$ correlations, potentially leading to structural errors.

Another way to show that 1,1-ADEQUATE is not quite a routine experiment compared to HMBC is the number of citations. The small number of papers utilizing 1,1-ADEQUATE (47 since the first publication in 1996) is in stark contrast with 8,500+ reports of the HMBC experiment (first publication is in 1986) uncovered in a parallel SciFinder search.

Considering the obvious S/N advantage of i-HMBC over 1,1-ADEQUATE and that cases where $^2J_{CH}$ is close to zero are uncommon, we recommend i-HMBC as the preferred method for identification of two-bond correlations reserving 1,1-ADEQUATE as a backup strategy if i-HMBC fails.

To address the comment regarding Figure 3a: our method measures relative isotope shift, which compares the HMBC chemical shifts from a specific proton to the most downfield HMBC chemical shift (reference) of that proton. Please note even longer-range HMBC signals still have a discernable isotope shift. Therefore, the relative isotope shift value measured by this method will be affected by the absolute isotope shift of the reference, which can be different between different protons. In addition, the isotope shifts will be affected by the chemical environment such as the hybridization state of the C-H. Thus, the relative isotope shifts are not directly comparable between different protons.

For HMBC signals *from the same proton*, in all of compounds we have investigated (~250 HMBC correlations) without exception, two-bond HMBC correlations have larger isotope shifts than

corresponding three-bond and longer range HMBC peaks. A more thorough study of the magnitude of the isotope shifts with theoretical calculations will be investigated in follow-up papers. The manuscript has been revised to add this discussion.

2) Will the broad readership and scope of this journal appreciate the work and its presentation?

The manuscript contains many technical details and descriptions about preliminary validation experiments, something the majority of the readers probably will not be interested in.

In conclusion, I find the research interesting and valid but recommend publication in a journal with a larger fraction of NMR spectroscopist readers.

We think the i-HMBC methodology presents a novel and superior workflow of NMR structure elucidation compared to traditional approaches. In addition to NMR spectroscopists, we envision that i-HMBC could be of great interest to broad readership of *Nature Communications*, particularly for the natural product chemistry and synthetic chemistry communities who regularly work with highly complex small molecules but are not familiar with in-depth NMR technologies and setup. Thus, considerable technical details are included for better comprehension and reproduction of our results and implementation of the i-HMBC experiment by non-NMR experts. Submitting to a specialized NMR journal would undermine the purpose of putting this technique before those in the natural products community, for example, who will benefit most by another powerful technique being available for challenging structure elucidation problems. In addition, we think that ${}^n\Delta^1\text{H}({}^{13/12}\text{C})$ isotope shift measurement methodology described in this manuscript is superior to the existing state-of-the-art method that was also published in *Nat. Commun.* (reference 38 in the manuscript). Given both the utility of these isotope shifts in organic chemistry and challenges associated with their measurements, we believe this manuscript will be of significant interest to a broader readership of *Nature Communications*.

Minor points:

- Inclusion of the values of the $nJ(\text{CH})$ coupling constants involved might be helpful and correlate with the isotope effects.

The origin of the isotope effect is related to anharmonic vibrations of chemical bonds in the molecule. NMR inactive nuclei would generate isotope effects as well, and these effects do not correlate with the magnitude of ${}^nJ_{\text{CH}}$, which originates from the scalar coupling interaction between two NMR active nuclei.

- How do the authors deal with situations without meaningful isotope shift differences, like when a proton shows only one correlation or the case with e.g. two two-bond correlations and nothing else? Measurement of individual isotope effects is referred to as challenging in line 168.

We thank the reviewer for this question and want to use this as an opportunity for additional clarification. The main utility of i-HMBC is distinguishing between 2-bond and longer-range HMBC correlations to facilitate structure elucidation by reducing the ambiguity of structural interpretation of HMBC correlations. A typical issue with HMBC data interpretation is having numerous correlations that could not be unambiguously assigned as 2-, 3- or 4-bond *a priori*. Our manuscript describes several high-profile examples (homodimericin A and B, cryptospirolepine and calicheamicin) where such complexity proved impossible to overcome without the orthogonal information such as 1,1-ADEQUATE, anisotropic

NMR data, or synthetic organic chemistry knowledge. In this manuscript, we demonstrate how structural ambiguity is resolved in all three of these cases by using i-HMBC to unambiguously distinguish 2-bond from longer-range HMBC correlations.

We agree that in a hypothetical case where there is only one HMBC correlation observed it would be impossible to measure the relative isotope shift. However, we'd argue that in this rare situation the isotope shift information is consequently unnecessary since having only one HMBC correlation makes structural interpretation rather straightforward. For example, determining the location of acetylation, which would only have a single HMBC correlation from the methyl to carbonyl group.

To further illustrate this point, all observed HMBC and LR-HSQMBC correlations of homodimericin A are shown below. Correlations observed in the 8 Hz optimized HMBC spectrum (shown in the left panel) of homodimericin A that could be assigned only *after* the structure was elucidated from 1,1-HD-ADEQUATE data. Without i-HMBC there are no means to differentiate $^2J_{CH}$ from $^3J_{CH}$ or even $^4J_{CH}$, and in a rigid structure, with fixed bond angles, a deleteriously complex coupling network exists that impedes structure elucidation. The situation is further exacerbated if one tries to resort to, for example, a 2 Hz optimized LR-HSQMBC spectrum. The correlations shown in the right panel are those that were observed *in addition* to the correlations that were observed in the 8 Hz HMBC spectrum. Practically speaking, the only segment of the structure in which the conventional HMBC data can be interpreted in a meaningful fashion is in the "tail" of the molecule. These difficulties confounded Professor Clardy and colleagues for three years and despite the available data being fed into a Computer-Assisted Structure Elucidation (CASE-ACD StructureElucidator™) program for 15 different computational runs, it was not until unambiguous assignments of 2-bond HMBC correlations (derived from 1,1-ADEQUATE data) were added to the data input file that the program was able to position the correct structure first in the output file from the computation.

Another example of the complexity of HMBC data that underscores the difficulty of interpreting those data is provided by strychnine. The structure shown below is a compilation of all of the long-range heteronuclear correlations that have been observed in the HMBC (8 and 4 Hz optimized) and LR-HSQMBC (2 Hz optimized) spectra. Weak correlations are denoted by dashed arrows; there are in excess of 160 long-range heteronuclear correlations for strychnine! A significant percentage of the correlations would not have been attributable if the structure was unknown.

- In fragments with long chains of quaternary carbons only the INADEQUATE experiment will be useful, something that deserves to be mentioned.

i-HMBC methodology developed in this work is intended to distinguish 2-bond from longer-range HMBC correlations. Of course, this is predicated on observing HMBC correlations to begin with. If the chain of quaternary carbons is long enough that only INADEQUATE experiment could establish the connectivities between those carbons, then, naturally, neither i-HMBC nor conventional HMBC or 1,1-ADEQUATE would be useful. It should be noted that INADEQUATE is used very rarely in practice: its sensitivity is too low and corresponding sample requirements are too high even with the use of modern cryoprobes. Sensitivity of 1,1-ADEQUATE was about 40x lower than that of i-HMBC on a modern, carbon-enhanced TCI cryoprobe (see Table S11). INADEQUATE sensitivity, since it is a ^{13}C -detected experiment, would be at least an order of magnitude less sensitive than 1,1-ADEQUATE, which translates to 2 orders of magnitude longer acquisition time.

- Have the authors considered other experiments than HMBC for measurement of the isotope effects of interest?

We thank the reviewer for this insightful suggestion. Indeed, the relative isotope shift could in principle be measured by any heteronuclear correlation experiments that are capable of yielding non-distorted line shape, enough ^1H dimension resolution and sufficient S/N. We have had success using HMQC (D-i-HMBC in this manuscript), HSQMBC and dual-optimized, inverted $^1J_{\text{CC}}$ 1,n-ADEQUATE. The manuscript has been updated to add the discussion.

Recommendations to the authors:

- Work on putting the handling of the overlap between two- and three-bond isotope effects on more solid ground. It is a good start that your procedure seems to correctly predict the number of intervening bonds on the molecules studied but more is needed to claim general applicability of the method.
- Work out how you will deal with the situations where there are no meaningful isotope shift differences but only individual isotope shifts.
- Discuss the possible use of other experiments than HMBC for measurement of these isotope effects.

We thank the reviewer for summarizing the recommendations. All action items have been addressed above in the responses to individual comments. The manuscript and the SI have been correspondingly updated.

REVIEWERS' COMMENTS

Reviewer #1 (Remarks to the Author):

Authors have successfully answered/considered all my suggestions and doubts. I think the article meets all criteria to be accepted for publication in Nat Comm. The proposed experiment is original and I'm sure it will be a powerful tool for structure elucidation studies of challenging chemical compounds and natural products, and an important reference for NMR spectroscopists.

Reviewer #3 (Remarks to the Author):

The method presented by the authors was clear in the first submission, and there are no surprises in the rebuttal and the revised manuscript that make me change the recommendation. That said, I am looking forward to follow the development and exploration of isotope shifts as a tool in NMR structure elucidation.

Various comments, mostly to the rebuttal document:

1,1 ADEQUATE is indeed a very simple experiment (sensitivity put aside) with not too many peaks in the spectrum. That said, it is true that one cannot blindly trust the presence of a peak indicating a one-bond correlation. That or something similar is the case for most NMR experiments relying on coherence transfer via J couplings. Unusual Js can lead to "unexpected" peaks and one must always beware of that.

I see no significance in the number of citations for 1,1 ADEQUATE versus HMBC. I presume the authors' recommendations would have been the same even if 1,1 ADEQUATE had had 10'000 citations. HMBC or similar is the standard first choice and there is no need to resort to a less sensitive experiment if it is not needed. Besides that, the majority of routine NMR users simply continue to use the techniques they practiced during their own studies even when they become aware of better alternatives.

They authors state that "cases where $2J(\text{CH})$ is close to zero are uncommon" which certainly depends on how "close" is defined. In ref. (7), several two-bond correlations were missing in the HMBC spectrum of strychnine recorded with a standard delay.

I get the authors' comments to my comment to Fig. 3a, which make me presume that two-bond correlations are needed to establish meaningful relative isotope shifts. If that is true, it would be good to have stated in the paper. If it is not true, it would be good to know what can be done in the absence of a two-bond correlation to a given proton.

It is an inaccurate statement to say that SEA XLOC "assumes that $2J(\text{CH})$ is negative". SEA XLOC clearly identifies heteronuclear long-range correlations where a passive negative J is involved, which helps the spectroscopist draw assignment conclusions.

Reviewer #1 (Remarks to the Author):

Authors have successfully answered/considered all my suggestions and doubts. I think the article meets all criteria to be accepted for publication in Nat Comm. The proposed experiment is original and I'm sure it will be a powerful tool for structure elucidation studies of challenging chemical compounds and natural products, and a important reference for NMR spectroscopists.

Thank you for the careful review and kind remarks.

Reviewer #3 (Remarks to the Author):

The method presented by the authors was clear in the first submission, and there are no surprises in the rebuttal and the revised manuscript that make me change the recommendation. That said, I am looking forward to follow the development and exploration of isotope shifts as a tool in NMR structure elucidation.

Thank you very much for your excellent additional feedback. We're also greatly looking forward to further developing this isotope shift methodology, including new pulse sequences and quantum chemical calculations, over the next several years.

Various comments, mostly to the rebuttal document: 1,1 ADEQUATE is indeed a very simple experiment (sensitivity put aside) with not too many peaks in the spectrum. That said, it is true that one cannot blindly trust the presence of a peak indicating a one-bond correlation. That or something similar is the case for most NMR experiments relying on coherence transfer via J couplings. Unusual Js can lead to "unexpected" peaks and one must always beware of that.

Yes, agreed.

I see no significance in the number of citations for 1,1 ADEQUATE versus HMBC. I presume the authors' recommendations would have been the same even if 1,1 ADEQUATE had had 10'000 citations. HMBC or similar is the standard first choice and there is no need to resort to a less sensitive experiment if it is not needed. Besides that, the majority of routine NMR users simply continue to use the techniques they practiced during their own studies even when they become aware of better alternatives.

Indeed, there is a significant challenge in convincing routine NMR users to implement and run new, advanced pulse sequences, such as SEA-XLOC. However, we see this as a primary advantage of i-HMBC because it only involves a minor modification to the nominal HMBC acquisition parameters, and thus we believe such familiarity will drive its utilization.

They authors state that "cases where $2J(\text{CH})$ is close to zero are uncommon" which certainly depends on how "close" is defined. In ref. (7), several two-bond correlations were missing in the HMBC spectrum of strychnine recorded with a standard delay.

Proper ${}^nJ_{\text{CH}}$ optimization is an important consideration when acquiring HMBC data, particularly for newly discovered, bioactive natural products. It is a common practice to run HMBC with high and low optimization (such as 8 Hz and 3 Hz) in order to observe correlations both through strong and weak couplings. i-HMBC being a mild modification of regular HMBC, can be used in the similar fashion if needed. And in the case of ref. (7), separate HMBC spectra with optimizations of 8 and 3 Hz would have provided nearly all two-bond correlations.

I get the authors' comments to my comment to Fig. 3a, which make me presume that two-bond

correlations are needed to establish meaningful relative isotope shifts. If that is true, it would be good to have stated in the paper. If it is not true, it would be good to know what can be done in the absence of a two-bond correlation to a given proton.

Actually this is not the case, and this is why the example of caffeine was shown in Fig. 3 in the manuscript. Caffeine only exhibits three-bond and longer range HMBC correlations. As seen in the figure, all isotope shift differences for caffeine fall within the range of 0 to -0.3 ppb, and thus they are properly classified.

It is an inaccurate statement to say that SEA XLOC "assumes that $2J(\text{CH})$ is negative". SEA XLOC clearly identifies heteronuclear long-range correlations where a passive negative J is involved, which helps the spectroscopist draw assignment conclusions.

Indeed, as mentioned in this comment SEA XLOC makes the distinction explicitly utilizing the sign of passive J , with the understanding that $2J_{\text{CH}}$ is negative in most cases and $3J_{\text{CH}}$ is always positive. This is explained in the original publication: "Three-bond J_{HH} and J_{CH} coupling constants are invariably positive and the two-bond J_{CH} coupling constant is in most cases negative. That can be used for distinction via states of heteronuclear zero- and double-quantum coherences (ZQC and 2QC, respectively) that are doublets in the above C–H2–H3 three-spin system." (Chem. Commun. 2018, 54, p. 9781)

In addition, SEA XLOC is based on heteronuclear J -couplings, while i-HMBC represents an entirely orthogonal approach based on isotope shifts, thus providing additional valuable structural information.